# Timing of Measles, Mumps, and Rubella Vaccination: Secondary Outcomes from an Immunological Survey

**DOI:** 10.3390/vaccines13040382

**Published:** 2025-04-03

**Authors:** Jana Zibolenová, Romana Ulbrichtová, Eva Malobická, Martin Novák, Tibor Baška, Lucia Časnocha Lúčanová, Ján Mikas, Adriana Mečochová, Henrieta Hudečková

**Affiliations:** 1Department of Public Health, Jessenius Faculty of Medicine in Martin, Comenius University in Bratislava, Malá Hora 4B, 036 01 Martin, Slovakia; jana.zibolenova@uniba.sk (J.Z.); romana.ulbrichtova@uniba.sk (R.U.); eva.malobicka@uniba.sk (E.M.); tibor.baska@uniba.sk (T.B.); henrieta.hudeckova@uniba.sk (H.H.); 2Neonatology Department, Jessenius Faculty of Medicine in Martin, Comenius University in Bratislava, University Hospital Martin, Kollárova 2, 036 59 Martin, Slovakia; lucia.casnochalucanova@uniba.sk; 3Faculty of Public Health, Slovak Medical University in Bratislava, Limbová 14, 833 03 Bratislava, Slovakia; jan.mikas@szu.sk; 4Public Health Authority of the Slovak Republic, Trnavská cesta 52, 826 45 Bratislava, Slovakia; adriana.mecochova@uvzsr.sk

**Keywords:** timing of vaccination, MMR vaccine, vaccination schedule, measles, immunological survey

## Abstract

**Background/Objectives:** This study analyzed data on the actual timing of the first and second doses of the Measles, Mumps, and Rubella (MMR) vaccination in Slovakia according to the vaccination schedule. **Methods:** Histograms were constructed using immunological survey data on MMR vaccination conducted in Slovakia in 2018. **Results:** For the first dose (2560 individuals), 83.4% of them were vaccinated timely (15th–18th month, mostly in the 16th month), while 13.8% of them were delayed. For the second dose (1061 individuals), 72.7% of vaccinations were timely (11th year), and 23.2% were delayed. There was a bimodal distribution of the timing of the administration of the second dose, with peaks at the beginning of the 11th year and at the turn of the 11th and 12th year. **Conclusions:** The unexpected shape of the histograms suggests that ambiguous interpretations of the vaccination schedule may be one of the causes of vaccination delays.

## 1. Introduction

Vaccination is one of the most important achievements of modern medicine, saving millions of lives yearly. However, recent years have shown a growing trend of vaccine hesitancy and declining vaccination rates in numerous countries, including Slovakia [1,2,3,4,5]. Additionally, a delay in administering individual doses against the recommended schedule is sometimes observed. This delay can be due to objective reasons, such as temporary contraindications (e.g., a child’s illness), or parental postponement.

The timely administration of the first dose of the MMR (Measles, Mumps, Rubella) vaccine is particularly important to minimize the duration of a child’s susceptibility to infection [6]. The duration of maternal immunity against measles is shorter in children of vaccinated mothers compared to those who have had the infection [7]. Since the vast majority of mothers today have not been exposed to the virus naturally, their children are protected by maternal antibodies for a shorter period than in the past, and therefore the gap between the disappearance of maternal antibodies and the first dose of the vaccine should be minimized as much as possible [8,9]. Adhering to the timing of the second dose is not as critical as for the first dose, as it is only considered a dose of certainty. While recommendations for the first dose are similar across European countries (typically between 12 and 18 months), there are significant differences across countries regarding the second dose. The European Centre for Disease Prevention and Control (ECDC) provides a summary of information via its Vaccination Scheduler regarding the specific ages for MMR vaccine administration among different European countries [10].

The immunological survey was conducted in Slovakia in 2018 [11]. Although its primary aim was to obtain an overview of the prevalence and dynamics of antibodies against selected infectious diseases, it also provided valuable information on the exact timing of vaccination.

The objective of this study was to summarize data on the actual timing of the first and second doses of the MMR vaccine in Slovakia and to analyze them against the recommended vaccination schedule. The findings of this study can contribute to the identification of possible weak points in the implementation of the schedule and thus to improve it.

## 2. Materials and Methods

This study has a descriptive, cross-sectional design and is related to the 2018 Immunological Survey (IS 2018), coordinated by the Public Health Authority of the Slovak Republic. The serosurvey was conducted during the summer of 2018 and involved 4218 participants aged 1–69 across Slovakia, selected using stratified sampling to ensure even distribution by gender, age, and region. All details regarding the procedures, ethical issues, and results associated with the implementation of IS 2018 are available in a separate publication [11].

The following vaccination schedule was in effect for the MMR vaccine during the survey period: first dose of the trivalent vaccine in the 15th–18th month (equivalent to 14–17 months), second dose in the 11th year (equivalent to 10 years). This vaccination schedule has been in force with only a few minor modifications since 1995 and is currently based on Decree No. 585/2008 Coll. of the Ministry of Health of the Slovak Republic [12]. Nowadays, the second dose is given at age 4 (equivalent to 5th year).

To determine the timing of the first/second dose of the MMR vaccine, a subgroup of individuals with a known date of birth after 1995 and date of vaccination was selected from the entire IS 2018 sample. Records with missing dates of vaccination and incorrect ones were excluded from the analysis. Age at the first/second dose vaccination was calculated as the difference between the date of vaccination and the date of birth.

Histograms showing the age distribution of the first and second doses were created using MS Excel. Since the time range for the first and second doses is significantly different, different interval widths were also chosen for display. For the first dose, the interval width was one day and the histogram was smoothed using a 7-day moving average. For the second dose, the interval width was one month, without smoothing. The histograms also display the distribution by year of birth.

Differences between ages at the first and second doses, stratified by sex and year of birth, aggregated into 5-year intervals (except for the last interval), were compared using the Kruskal–Wallis test followed by post-hoc analysis with Bonferroni correction. The analysis was performed using SPSS Statistics for Windows, Version 29.0 (IBM Corp., Armonk, NY, USA).

## 3. Results

### 3.1. First Dose

A total of 2713 individuals from IS 2018 were born after 1995 and in 2560 of them, the age at first dose of vaccination was available in the records. Details on the availability of vaccination records are provided in Table 1.

The first dose was administered in accordance with the vaccination schedule (between the 15th and 18th month) in 2137 individuals (83%), with a significant predominance of administration in the 16th month; in 353 individuals (14%), the administration was delayed (Appendix A, Figure 1a). The age at the first dose of vaccination does not vary with sex or year of birth (*p* > 0.05).

### 3.2. Second Dose

A total of 1061 individuals born after 1995 had available records on the second dose of MMR vaccine. It was administered in accordance with the vaccination schedule (i.e., in the 11th year) in 771 individuals (72.7%), and in 246 individuals (23.2%), the administration was delayed.

The histogram (Figure 1b) shows a bimodal distribution of the timing of the second dose with two peaks: the first at the beginning of the 11th year and the second at the turn of the 11th and 12th year.

Significant differences in the timing of the second dose were observed in relation to the year of birth (*p* < 0.001). As the year of birth increased, the proportion of later administrations decreased, and for those born between 1998 and 2006, the proportion was relatively stable (Appendix A). The birth cohorts of 2007 and 2008 were vaccinated with the second dose during the IS 2018 period, therefore late administrations could not be identified.

## 4. Discussion

Children who have not yet been vaccinated rank among the most vulnerable groups. Thus, delaying the first dose of vaccination prolongs the period during which the child is not protected. The highest age-specific incidence of measles in Europe is among 0-year-olds, followed by 1- to 4-year-olds [13,14].

Slovakia has successfully prevented large-scale measles epidemics for a long time, essentially since the 1990s, thanks to its high vaccination rate. Between 1998 and 2018, only 24 cases (imported or import-related) were reported.

The good epidemiological situation was interrupted only in 2018/2019 in Roma communities in eastern Slovakia. During this outbreak, nearly all age groups were affected, excluding the oldest. The epidemiological investigations revealed long-term inconsistencies of vaccination records within these marginalized communities, leading to an emergence of susceptible unvaccinated groups of individuals, which resulted in a local epidemic outbreak. Unvaccinated children under the age of the first dose were, of course, among the most severely affected. However, in recent years, we have observed an overall decline in vaccination rates, which in the future may lead to further outbreaks [5].

Due to a previous long-lasting favorable epidemiological situation, Slovakia ranks among European countries where the first dose of MMR vaccination is recommended only after the 14th month. It is later than in most developed countries [10,15]. In an outbreak, it is possible to administer the vaccine to 6- to 11-month-old children within 3 days of their last contact with a measles patient. Vaccination is key to preventing the spread of the disease; however, this dose is considered a zero dose [16].

The constructed histograms show an uneven distribution of vaccination age for both the first and second doses of the MMR vaccine. It seems as if there is a problem with interpreting the immunization schedule. Although the first dose is recommended from the 1st day of the 15th month of life (in other words, the first day after reaching 14 months of age), the vast majority of children are vaccinated in the first days of the 16th month of life. We think that some healthcare personnel interpret the 15th month as 15 months and therefore actually vaccinate children in their 16th month. It is not possible to exactly identify the cause of this shift from the analyzed data. The reasons for vaccination delays can include numerous reasons, including short-term outages in vaccine supply, logistical problems with medical appointments, temporary contraindications (particularly acute respiratory illness), and unjustified delays and hesitations by parents [17,18,19,20]. Considering the sharp increase in vaccinations precisely at the 16th month of life, we assume that the delay is most likely associated with an ambiguous interpretation of the immunization schedule by healthcare personnel.

This observed delay in administering the first vaccination dose, although still in accordance with the vaccination schedule, can present a public health issue. Such a shift can significantly increase the number of susceptible children in the population and should be a reason for concern [21]. Further investigations are needed to identify the main reasons for these delays and the related factors. Based on these results, effective interventions can be proposed to decrease the risk of the spread of infection among vulnerable populations of children not yet vaccinated.

In the case of the second dose, a bimodal distribution of the age of administration is unexpectedly observed in the histogram. These two maxima again indicate that there may be an inaccurate/dual interpretation of the vaccination schedule: although this vaccination can be administered as early as from the beginning of the 11th year, a significant proportion of pediatricians do it only towards the end of the 11th year. However, the issue needs further investigation, as other factors such as logistical problems, vaccination hesitancy, and temporary contraindications may also contribute to vaccination delays.

The COVID-19 pandemic has changed the overall perception of vaccination. During the pandemic, the estimated global coverage of the first dose of MMR declined [22]. Unfortunately, this dataset covers only the pre-pandemic period, which is a limitation. While this study does not deal with hesitancy, public perception of vaccination, or vaccination coverage, it highlights a single, previously unknown aspect: the dual interpretation of the vaccination schedule, which appears to be independent of the COVID-19 pandemic.

Since measles remains a pressing global public health issue, and many developed countries face similar challenges in their vaccination programs, our findings have noteworthy international significance [23].

## 5. Conclusions

Although we observed a shift in the timing of the first dose of the MMR vaccine of only one month, this delay could endanger the health of a significant number of children. Since unvaccinated children are the most at-risk group for the spread of measles, it is essential to minimize the risk and keep the group of unvaccinated children as small as possible. One way to achieve this is by strictly adhering to the immunization schedule, ensuring that the first dose is administered at the earliest recommended time. Therefore, it is necessary to address the correct interpretation of the immunization schedule and to educate the public and healthcare professionals about the importance of the correct timing of mandatory vaccination. This means avoiding unnecessary delays and ensuring that the vaccine is administered as soon as possible.

## Figures and Tables

**Figure 1 vaccines-13-00382-f001:**
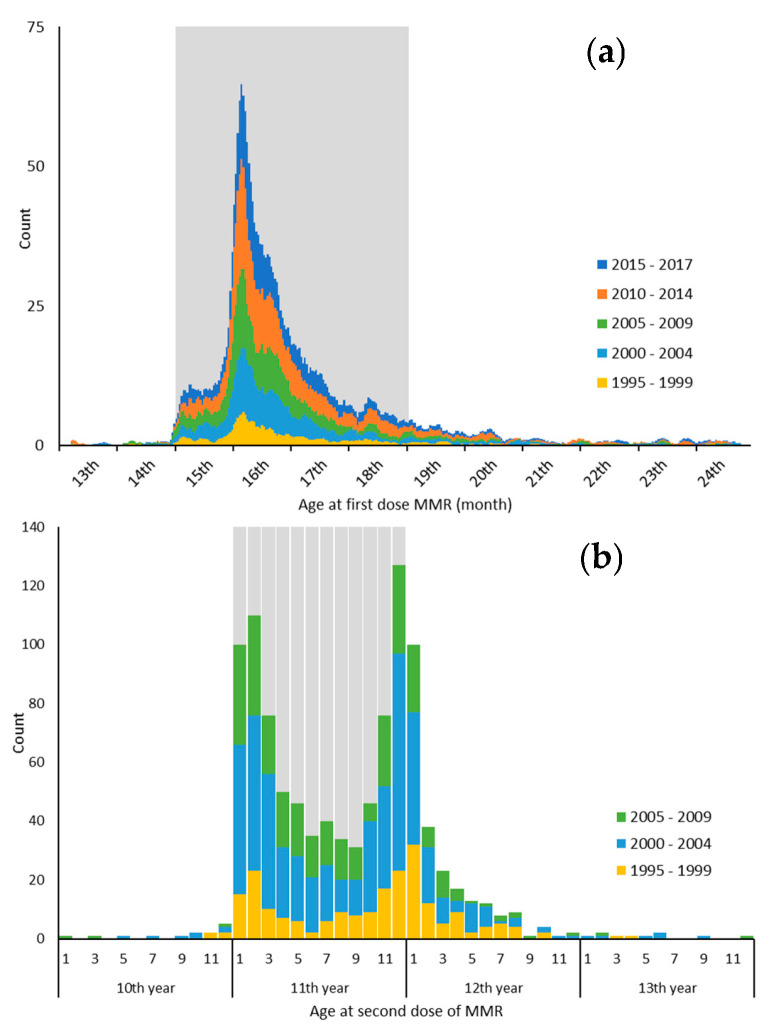
Histograms of age at first (**a**) and second (**b**) dose of MMR (Measles, Mumps, and Rubella) vaccine, split according to the year of birth. Grey zone indicates the vaccination schedule. Immunological Survey 2018, Slovakia.

**Table 1 vaccines-13-00382-t001:** Details on the availability of vaccination records from Immunological Survey 2018, Slovakia (persons born after 1995).

Total (Birth Cohorts 1995–2018)	2713	100.0%
1st dose		
Unvaccinated due to age (less than 15th month)	13	0.5%
Unvaccinated (15th–18th month)	28	1.0%
Unvaccinated (more than 18th month)	24	0.9%
Unknown vaccination status for 1st dose	18	0.7%
Vaccinated, unknown date	70	2.6%
Vaccinated, known date	2560	94.4%
2nd dose		
Unvaccinated due to age (less than 11th year)	1469	54.1%
Unvaccinated (11th year)	71	2.6%
Unvaccinated (more than 11th year)	NA	NA
Unknown vaccination status for 2nd dose	55	2.0%
Vaccinated, unknown date	57	2.1%
Vaccinated, known date	1061	39.1%

NA—not available.

## Data Availability

The dataset is available upon request from the authors.

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
