# Peer review of "Timing of Measles, Mumps, and Rubella Vaccination: Secondary Outcomes from an Immunological Survey"

_vaccines, 2025, doi:10.3390/vaccines13040382_

Round 1

Reviewer 1 Report

Comments and Suggestions for Authors

Summary: Authors analyzed the data on the actual timing of the first and second doses of the Measles, Mumps and Rubella (MMR) vaccination in Slovakia.

Major comments:

  1. It would be great if authors discuss about the rate of measles in Slovakia in the past 10 years and perform some correlation analyses with vaccination rate.
  2. A figure which compares the age of first dose vaccination among different European countries would be interesting as well.

Author Response

We thank the Reviewer 1 for their comments. Below are our responses to their comments.

Comment 1: It would be great if authors discuss about the rate of measles in Slovakia in the past 10 years and perform some correlation analyses with vaccination rate.

Response 1: We added a paragraph to the Discussion about the recent measles outbreak and vaccination coverage in Slovakia.

Comment 2: A figure which compares the age of first dose vaccination among different European countries would be interesting as well.

Response 2: We added information regarding ECDC vaccination Scheduler also with appropriate citation into the introduction section: The European Centre for Disease Prevention and Control (ECDC) provides a summary of information via its Vaccination Scheduler regarding the specific ages for MMR vaccine administration among different European countries [10].

Reviewer 2 Report

Comments and Suggestions for Authors

I appreciate the invitation to review this paper. My main background in is biostatistics with a focus on healthcare related populational data assessments. I have experience working in vaccination  and vaccine hesitancy projects, this paper is a perfect for my expertise. I believe it is important to disclose that I am American, and my context is tied to the American experience. This may be relevant in one of my concerns.

The paper is well written with a straightforward narrative. I did not detect any issues with the English language used. The paper is interesting as it takes advantage of previously collected data but with a different context.

There are two major concerns I would recommend the authors to address or comment.

First, I have some concern with the narrative presented through the main arguments on how the schemes are likely being inaccurately or dually interpreted. These arguments are based on the vaccination occurring more often on the 16th month rather the 15th month and a similar situation that occurs for the second dose. However, could this be a matter of logistics. How easy it is for parents to get an appointment (if those are necessary for vaccination) to match the recommended time. If there is a delay with scheduling, this would explain the delay in vaccination. For second doses, based on my American experience, vaccines are administered within the yearly visits. If there is low adherence to yearly visits and some visits are skipped, this would explain why they are given on the next yearly visit. I believe this whole possibility may have been ignored in the narrative.

Second, the study uses a cross-sectional dataset from 2018, this is pre-COVID. Vaccine hesitancy in America changed because of COVID, I am not sure if this occurred in Slovakia as well. If that is the case, how significant is this study considering that it is pre-COVID, and it may not be representative anymore? I suggest being more upfront by mentioning this concern in the limitations.

A minor comment on Figure 1b. Using the bars and placing each year next to the other, gives an impression of continuity, this is confusing. Each year has its own month distribution, I believe this can be fixed by adding some space between each year.

I believe all my concerns can be resolved though some revision. I am looking forward to a revised version of the paper.

Author Response

We thank the Reviewer 2 for their comments. Below are our responses to their comments. 

Comment 2: 

I appreciate the invitation to review this paper. My main background in is biostatistics with a focus on healthcare related populational data assessments. I have experience working in vaccination and vaccine hesitancy projects, this paper is a perfect for my expertise. I believe it is important to disclose that I am American, and my context is tied to the American experience. This may be relevant in one of my concerns.

The paper is well written with a straightforward narrative. I did not detect any issues with the English language used. The paper is interesting as it takes advantage of previously collected data but with a different context.

There are two major concerns I would recommend the authors to address or comment.

First, I have some concern with the narrative presented through the main arguments on how the schemes are likely being inaccurately or dually interpreted. These arguments are based on the vaccination occurring more often on the 16th month rather the 15th month and a similar situation that occurs for the second dose. However, could this be a matter of logistics. How easy it is for parents to get an appointment (if those are necessary for vaccination) to match the recommended time. If there is a delay with scheduling, this would explain the delay in vaccination. For second doses, based on my American experience, vaccines are administered within the yearly visits. If there is low adherence to yearly visits and some visits are skipped, this would explain why they are given on the next yearly visit. I believe this whole possibility may have been ignored in the narrative.

Response 1: 

We agree that vaccination postponement should include numerous reasons, and one of them should be a logistical problem in obtaining a medical appointment. So we added this reason to the list in the discussion.

With the second dose, in Slovakia, medical preventive check-ups are scheduled every two years between the ages of 3 and 18 years and parents are proactivelly invited by respective pediatricians. If the visit is skipped, pediatricians do not wait two years for another visit but call the parents. Of course, some parents postpone and ignore the visits, but we assume that this reason would not result in a bimodal distribution of the age of second dose in Slovakia's case.

Comment 2: 

Second, the study uses a cross-sectional dataset from 2018, this is pre-COVID. Vaccine hesitancy in America changed because of COVID, I am not sure if this occurred in Slovakia as well. If that is the case, how significant is this study considering that it is pre-COVID, and it may not be representative anymore? I suggest being more upfront by mentioning this concern in the limitations.

Response 2:

We agree with the reviewer; the COVID-19 pandemic changed how vaccination is perceived, and vaccination hesitancy has also changed in Slovakia. We added a short paragraph regarding the COVID-19 pandemic to the discussion. However, this manuscript does not deal with hesitancy and public perception of vaccination, but highlights a single, previously unknown aspect: the dual interpretation of the vaccination schedule, which we thought is independent of the COVID-19 pandemic.

Comment 3: 

A minor comment on Figure 1b. Using the bars and placing each year next to the other, gives an impression of continuity, this is confusing. Each year has its own month distribution, I believe this can be fixed by adding some space between each year.

Reponse 3: We added spaces between bars.

I believe all my concerns can be resolved though some revision. I am looking forward to a revised version of the paper.